# Community-based health insurance, healthcare service utilization and associated factors in South Gondar Zone Northwest, Ethiopia, 2021: A comparative cross-sectional study

**Fentaw Teshome Dagnaw**[1]*, **Melkalem Mamuye Azanaw**[1], **Aytenew Adamu**[2], **Tinsaea Ashagrie**[2], **Abdelah Alifnur Mohammed**[2], **Hiwot Yisak Dawid**[1], **Mulu Tiruneh**[1], **Biruk Demissie**[1], **Getaneh Atikilt Yemata**[1], **Getachew Yideg Yitbarek**[3], **Yikeber Abebaw**[4], **Habtamu Shimels Hailemeskel**[5]

**1** Department of Public Health, College of Health Sciences, Debre Tabor University, Debre Tabor, Ethiopia, **2** Department of Sociology, College of Social Sciences and Humanities, Debre Tabor University, Debre Tabor, Ethiopia, **3** Department of Biomedical Sciences, College of Health Sciences, Debre Tabor University, Debre Tabor, Ethiopia, **4** Department of Statistics, College of Natural and Computational Sciences, Debre Tabor University, Debre Tabor, Ethiopia, **5** Department of Paediatrics and Neonatal Nursing, College of Health Sciences, Debre Tabor University, Debre Tabor, Ethiopia

* fentawtesh6@gmail.com

## Abstract

### Introduction

Community-based health insurance schemes are becoming increasingly recognized as a potential strategy to achieve universal health coverage in developing countries. Ethiopia has implemented community-based health insurance in piloted regions of the country. The scheme aims to improve the utilization of healthcare services by removing financial barriers. There is a dearth of literature regarding the effect of the insurance scheme on the utilization of healthcare services.

### Methods

A community-based comparative cross-sectional study was conducted in the south Gondar Zone. Six hundred fifty-eight participants were selected using a systematic random sampling method. Data were entered into EPI data version 4.4.1 and exported to SPSS version 25 for analysis. Binary logistic regression was used to measure the association of factors with the outcome variable. The result of the final model was expressed in terms of Adjusted Odd Ratios (AOR) and 95% CI.

### Result

Two hundred twenty-three (67.8%) and 111 (33.7%) of the respondents reported that their family members went to health institutions within three months among CBHI users and non-users respectively. The presence of under-five children (AOR = 2, 95% CI = 1.6–2.4), CBHI

**Data Availability Statement:** All relevant data are within the paper and its Supporting information files.

**Funding:** The author(s) received no specific funding for this work.

**Competing interests:** The authors have declared that no competing interests exist.

**Abbreviations: CBHI**, Community Based Health Insurance; **CHPS**, Community-based Health Planning and Services; **CI**, Confidence Interval; **DTU**, Debre Tabor university; **SDGs**, Sustainable Development Goals; **SPH**, Social and Population Health; **SPSS**, Statistical Package for the Social Sciences; **UHC**, Universal Health Coverage; **WHO**, World Health Organization.

scheme membership times (AOR = 3, 95% CI = 2.6–3.4), household wealth index rich (AOR = 4, 95% CI = 2.3–6.3), household wealth index medium (AOR = 3, 95% CI = 1.8–5.8) and presence of chronic illness (AOR = 0.5, 95% CI = 0.2–0.8) was associated with health care service utilization. Households who were enrolled in CBHI were more likely to use healthcare services than households who were not enrolled.

## Conclusion and recommendation

Households who were enrolled in CBHI were more likely to use healthcare services than households who were not enrolled. Therefore, health sector leaders and managers in the study area should strengthen their efforts for increasing the enrollment of the community into CBHI.

## Introduction

Health insurance is a method of distributing the financial risk associated with the variation in individuals' health-care expenses by pooling costs over time through prepayment and across people through risk pooling [1]. Community-based health insurance is a non-profit organization that aims to increase financial access to health care services while also protecting its members from the financial risks that come with the disease. Its institutional arrangements are meant to maximize its core tasks of revenue collecting, risk-sharing, and purchasing of health care services, and it operates based on solidarity and mutual help values [2].

People use health-care services for a variety of reasons, including curing or treating illnesses and disorders, preventing or delaying future health problems, reducing pain and improving quality of life [3]. The numerous enabling elements must favorably converge for a healthcare consumer to make optimal use of the healthcare system. Several aspects of healthcare access, such as the availability of services, affordability, and transportation, as well as health literacy skills in communicating with healthcare providers and social support to facilitate decision-making, treatment adherence, and healthy behaviors, are among these factors [4].

Furthermore, for the lower socioeconomic groups of society, out-of-pocket medical expenses result in significant financial hurdles and a poor quality of life in the home. Approximately 44 million households around the world are experiencing financial hardship as a result of healthcare costs. As a result, each year, around 25 million households live in extreme poverty [5, 6]. All persons who require health services can receive them under Universal Health Coverage (UHC) without incurring undue financial hardship. As a result, universal health coverage (UHC) is a fundamental component of long-term development and poverty reduction, as well as a critical component of any endeavor to decrease social inequities and improve access to care [7].

According to a recent study, most civilizations have a sense of social solidarity when it comes to health-care access and prices, however, the nature and intensity of these attitudes varies by environment [6]. Since 2011, Ethiopia has been piloting Community-based Health Insurance (CBHI) schemes to learn from them and eventually scale them up across the country. It began with 13 district designs that have yielded promising preliminary findings [2, 8].

The use of modern health care services has remained quite low in most Low and Middle Income (LMIC) nations [9]. Out-of-pocket (OOP) expenditure on health care has significant implications for poverty in many developing countries [10]. Financial constraints were shown

to be one of the biggest hurdles to accessing and using contemporary healthcare services in low and middle-income nations, according to research. Furthermore, the lack of pre-payment financial arrangements for health care and the poor quality of healthcare services provided by public providers were key contributors to the reduced utilization of health care services [3, 10, 11].

In research conducted in Southern Ethiopia, CBHI scheme member families used much more health-care services for illness than non-member households [12]. Mechanisms to increase inclusion should be designed while socioeconomic characteristics remain key predictors of insurance participation. Improved community participation can increase community trust in the insurance and ultimately coverage [13]. According to a study conducted in North Achefer woreda, insured households in CBHI used health services more frequently. Educational status, family size, occupation, marital status, travel time to the nearest health institution, perceived quality of care, the first choice of location for treatment during illness, and expected healthcare cost of a recent treatment were all linked to health care utilization among CBHI users in this study [14].

Even though Ethiopia has been implementing the CBHI plan to promote the health of impoverished rural inhabitants since 2011, the majority of families are not enrolled, and there are few findings in the literature on the impact of CBHI on healthcare utilization in Ethiopia [2, 8]. The meaning of enrolment in CBHI and the link between CBHI with healthcare utilization are not well described. Hence, this study aimed to identify differences in enrolment in CBHI and to describe the link between CBHI enrolment and healthcare utilization and associated factors among South Gondar Zone population.

## Methods and materials

### Study design, area and period

The study was conducted in South Gondar Zone, Northwest Ethiopia, in 2012 E.C. South Gondar Zone is located in the Amhara region and its capital city Debre Tabor is 667km Northwest of Addis Ababa, the capital city of Ethiopia and 103 km to the southwest of Bahir dar. It has 21 weredas and 401 kebeles with a total population of 2,578,906 from this 1,276,558 are males and 1,302,348 are females. There are 8 hospitals and 96 health centers in the Zone. The study was conducted from October 2020—March 2021. A community-based comparative cross-sectional study was conducted.

### Source and study population

**Source population.** The source population for cases: All CBHI user households in the South Gondar Zone.

The source population for controls: CBHI non-user households in the South Gondar Zone.

**Study population.** For cases: Selected CBHI user households in the selected kebeles in the South Gondar Zone.

For controls: Selected CBHI non-user households in the selected kebeles in the South Gondar Zone.

### Variables

**Dependent variable.** Utilization of healthcare services.

**Independent variable. Socio-demographic variables**:—Age, religion, level of education, marital status, occupational status, family size, presence of children aged 0–5 years, presence of elderly above 60 years and household wealth index.

**Healthcare access and related factors**:—Current health status, presence of chronic illness, type of nearest health institutions, the first choice of treatment place, family members go to health institutes within three months, household enrolled in any other solidarity group and source of information for CBHI.

## Operational definitions

**Utilization of healthcare** was measured as the number of visits made by at least one household member at least once within 3 months for health services (diagnostic or treatment).

**Chronic illness** is a disease condition that lasts more than 3 months.

**Wealth index** was assessed by asking the following components of assets: livestock, crop production, infrastructure (radio, modern bed, mattress, phone, water pump, modern stove), latrine, housing condition (number of rooms, roof) and total farm size. The household wealth index was computed using principal component analysis. Although there were large data sets, principal component analysis is a technique to reduce the dimensionality of large data sets. The wealth index was categorized as poor, medium, and rich. The wealth index of the study households ranges from poor to rich [14].

**Households indexed in the muster book** of the CBHI schemes were recruited as insured, while households that were not indexed to the CBHI schemes were recruited as uninsured. A muster book is a registration book that indicates whether a household is a member of a CBHI or not.

## Sample size determination and sampling technique

**Sample size determination.** The sample was estimated using two population proportion formulas by using Epi-info stat calc with the following assumptions: 80% statistical power with a level of significance at 5%, insured to the un-insured ratio of 1:1, and the proportion of health service utilization was 35% for the insured households and 20% for the uninsured household [8]. The calculated sample size was 303. With a tolerable non-response rate of 10%; and a design effect of 2 the resulting sample size was 668 households.

**Sampling procedure.** A multistage sampling procedure was employed. From South Gondar, Zone 3 woredas (Lay Gaynt, Addis Zemen, and Woreta) were selected from those six kebeles were randomly selected. A total of 668 households were proportionally allocated to each sampled kebeles depending on the insurance status. Taking a fresh list of CBHI scheme member households available at the Kebeles' administration office. Systematic sampling was used to select the study subjects for both CBHI members and the comparative non-member groups. The respondents were heads of households for both CBHI members and the comparative non-member groups.

## Data collection methods and procedures

Face-to-face interviews using a semi-structured questionnaire were used to collect the data. The tool was composed of socio-demographic, healthcare access-related factors and perceived health needs questions. The data collectors were health professionals who are working in South Gondar Zone Hospitals.

## Data management and analysis

Data was entered into the computer by using the Epi-data version 4.4.1 and exported to SPSS version 25 for analysis. Tables, range and frequency were used to summarize and present the descriptive statistics of the data. Binary logistic regression analysis was conducted to assess the

association between dependent and independent variables. Independent variables which show association in bi-variable logistic regression analysis and those which have a P-value less than 0.25 entered into the multivariable logistic regression model, to identify significant factors associated with outcome variables. Finally, significant factors were identified based on AOR with a 95% Confidence level by considering P-value less than 0.05.

### Data quality assurance

To maintain its consistency the English version of the questionnaire was translated to Amharic versions and then back to English. Before data, collection training was given to the data collectors for two days. The pre-test was performed at a site other than the study area for its completeness among 10% of the participants at Farta Woreda. The questionnaire was checked for its completeness on daily basis.

### Ethical consideration

Ethical approval from a research ethics committee of Debre Tabor University was obtained with the ethical reference number dtu/chs/1154/2013. Informed written consent was taken from all study participants. The confidentiality of information and privacy of participants during the interview was respected. A detailed explanation was given to the study participants about the objectives and benefits of the study. Confidentiality and anonymity of the respondent's information were kept.

## Results

### Sociodemographic characteristics

A total of 658 households participated in the study with a response rate of 98.5%. The mean age of the respondents was 39.3 (SD ± 10.8) years with a minimum and maximum age of 18 and 75 years respectively. Most of the respondents were currently married 516 (78.4%). Most of the respondents 606 (92.1%) were Orthodox Christian followers. Regarding the level of education 279 (42.4%) of the respondents were unable to read and write and 290 (44.1%) of them were able to read and write. The majority of the respondents 457 (69.5%) were farmers and half 339 (51.5%) have three to five family sizes. Regarding the household wealth index, 313 (47.6%) were poor, 191 (29%) were medium and 154 (23.4%) of the respondents were rich. About 63% of the respondents and only 14% of the respondents report that there was a presence of under-five aged children and above 60 years elders respectively (Table 1).

### Health care and access to related causes and community-based health insurance

Six hundred and six (92.1%) of the respondents are healthy regarding their current health status. Regarding their primary choice of respondents 390 (59.3%) choices health centers, 194 (29.5%) hospitals the rest are private clinics and holly water. There was no chronic illness in 491 (74.6%) of the respondents. Two hundred twenty-three (67.8%) and 111 (33.7%) of the respondents reported that their family members went to health institutions within three months of CBHI users and non-users respectively. Six hundred forty (97.3%) of the respondents reported that they have used other solidarity groups like ekub and edir. The source of information about CBHI was 316 (58.1%) from a person who previously used CBHI, 299 (55%) from kebele leaders and 158 (29%) from mass media (Table 2).

**Table 1. Sociodemographic characteristics of respondents in South Gondar Zone, Northwest Ethiopia, 2021 (n = 658).**

| Variables | Category | Frequency | Percentage (%) |
|---|---|---|---|
| Residence | Urban | 57 | 8.7 |
| | Rural | 601 | 91.3 |
| Age of respondents | 18–24 years | 52 | 7.9 |
| | 25–44 years | 387 | 58.8 |
| | 45–65 years | 211 | 32.1 |
| | >65 years | 8 | 1.2 |
| Marital Level | Currently married | 516 | 78.4 |
| | Currently unmarried | 142 | 21.6 |
| Religion | Muslim | 52 | 7.9 |
| | Orthodox Christian | 606 | 92.1 |
| Level of education | Cannot able to read and write | 279 | 42.4 |
| | Able to read and write | 290 | 44.1 |
| | Primary level (1–8 grades) | 40 | 6.1 |
| | Secondary (9–12) | 32 | 4.9 |
| | College and above | 17 | 2.6 |
| Employment status | Farmer | 457 | 69.5 |
| | Private work | 32 | 4.9 |
| | Merchant | 92 | 14 |
| | Student | 47 | 7.1 |
| | Housewife | 17 | 2.6 |
| | Others | 13 | 2 |
| Household wealth index | Poor | 313 | 47.6 |
| | Medium | 191 | 29 |
| | Rich | 154 | 23.4 |
| Presence of under-five children | Yes | 413 | 62.8 |
| | No | 245 | 37.2 |
| Presence of elders above 60 years | Yes | 93 | 14.1 |
| | No | 565 | 85.9 |

### Factors associated with health service utilization

The odds of having under-five year children in the household were 2 times (AOR = 2, 95% CI = 1.6–2.4) more likely to utilize health care when compared with those who hadn't under-five year children. The odds of CBHI users were 3 times (AOR = 3, 95% CI = 2.6–3.4) more likely to utilize health care when it is compared to CBHI non-users. The odds of rich people and medium people were 4 (AOR = 4, 95% CI = 2.3–6.3) and 3 (AOR = 3, 95% CI = 1.8–5.8) times more likely to utilize health care services when compared with poor people respectively. The odds of respondents who had a chronic illness were 0.5 (AOR = 0.5, 95% CI = 0.2–0.8) times more likely to utilize health services when compared to those who had no chronic illness (Table 3).

In the past 3 months before data collection, 223 (67.8%) households who were enrolled in CBHI utilized healthcare services, and 111 (33.7%) households who were non-enrolled in CBHI utilized healthcare services. The chi-square result showed that there is a significant difference in health care utilization among CBHI enrolled and Non-enrolled households ($\chi^2$ = 76.3, P<0.001) (Table 4).

**Table 2. Health care access and related causes and community-based health insurance of respondents in South Gondar Zone, Northwest Ethiopia, 2021 (n = 658).**

| Variables | Category | | Frequency | Percentage (%) |
|---|---|---|---|---|
| Nearest health institution | Health post | | 162 | 24.6 |
| | Health center | | 485 | 73.7 |
| | Hospital | | 11 | 1.7 |
| Current health status | Healthy | | 606 | 92.1 |
| | Not healthy | | 52 | 7.9 |
| Primary health care choice | Health center | | 390 | 59.3 |
| | Hospital | | 194 | 29.5 |
| | Private clinic | | 41 | 6.2 |
| | Holy water | | 33 | 5 |
| Presence of chronic illness | Yes | | 167 | 25.4 |
| | No | | 491 | 74.6 |
| Do family members go to health institutes within three months? | CBHI Users | Yes | 223 | 67.8 |
| | | No | 106 | 32.2 |
| | CBHI Non-users | Yes | 111 | 33.7 |
| | | No | 218 | 66.3 |
| Household enrolled in any other solidarity group | Yes | | 640 | 97.3 |
| | No | | 18 | 2.7 |
| Source of information for CBHI | Health professionals | | 66 | 12.1 |
| | Kebele leaders | | 299 | 55 |
| | From another person who previously uses | | 316 | 58.1 |
| | Mass media | | 22 | 4 |

## Discussion

This study aimed at assessing the level of community-based health insurance healthcare service utilization and associated factors in South Gondar Zone Northwest, Ethiopia. In our study households who were enrolled in CBHI were more likely to use healthcare services than households who were not enrolled. This is consistent with studies done in different parts of Ethiopia where enrolment in healthcare insurance increases the use of healthcare in various settings but, households have relied on out-of-pocket spending and health insurance coverage remains low [14–16]. A health insurance scheme is a crucial strategy for the financial protection of many households [17]. The findings in this study also proved the fact that health insurance enrolment significantly improved health service utilization in which 67.8% and 33.7% of the respondents reported that their family members went to health institutions within three months among CBHI users and non-users respectively. This finding was higher than to study findings in Burkina Faso where the percentage of healthcare utilization was 37% among insured and 12% among uninsured and in Southern Ethiopia [15, 18]. This might be due to the difference in socio-demographic characteristics.

In this study, the odds of having under-five year children in the household were 2 times more likely to utilize health care when compared with those who hadn't the under-five year. This may be due to the fact that under-five year children need more follow-up in health institutions like for having a vaccination. In our study, rich peoples and medium peoples were 4 and 3 times more likely to utilize health care services when compared with the odds of poor people respectively. This finding was consistent with studies done in Ghana, China and Ethiopia [17, 19–22]. This might be due to rich and medium people having little limitation in terms of money to pay for health service utilization. Respondents who had a chronic illness were 0.5

**Table 3. Factors associated with health care utilization of respondents in South Gondar Zone, Northwest Ethiopia, 2021 (n = 658).**

| Explanatory Variables | Category | Utilized health-care services | | Un-adjusted | | Adjusted | |
|---|---|---|---|---|---|---|---|
| | | Yes | No | OR | 95% CI | OR | 95% CI |
| Residence | Urban | 42 | 15 | 1 | | 1 | |
| | Rural | 418 | 183 | 1.22 | 1.05–1.39 | 1.12 | 0.99–1.11 |
| Marital status | Married | 361 | 155 | 1 | | 1 | |
| | Not married | 99 | 43 | 1.01 | 0.93–1.09 | 0.93 | 0.83–1.3 |
| Presence of under-five children | No | 194 | 51 | 1 | | 1 | |
| | Yes | 266 | 147 | 2.1 | 1.6–2.6 | 2 | 1.6–2.4** |
| Presence of elders >60 years | Yes | 77 | 16 | 1 | | 1 | |
| | No | 383 | 182 | 2.3 | 0.97–3.9 | 1.8 | 0.99–2.7 |
| CBHI scheme membership | Users | 223 | 106 | 1 | | 1 | |
| | Non-users | 111 | 218 | 4.1 | 2.9–5.7 | 3 | 2.6–3.4* |
| Household wealth index | Poor | 250 | 63 | 1 | | 1 | |
| | Medium | 90 | 101 | 4.4 | 2.2–6.2 | 3 | 1.8–5.8* |
| | Rich | 50 | 104 | 8.3 | 4.6–12.6 | 4 | 2.3–6.3* |
| Current health status | Not healthy | 47 | 5 | 1 | | 1 | |
| | Healthy | 413 | 193 | 4.4 | 1.2–7.6 | 1.8 | 0.98–2.62 |
| Chronic illness | No | 329 | 162 | 1 | | 1 | |
| | Yes | 131 | 36 | 0.6 | 0.3–0.9 | 0.5 | 0.2–0.8** |

* = P-value < 0.001,

** = P-value < 0.05,

1 = reference category, OR = Odds Ratio CI = Confidence interval

(AOR = 0.5, 95% CI = 0.2–0.8) times more likely to utilize health services when we compared with respondents who had not chronic illness. This finding was consistent with studies done in Ghana, China and Ethiopia [12, 16, 17, 19]. This may be due to people with chronic illness having visited health institutions for consecutive followup.

## Conclusion and recommendations

Two hundred twenty-three (67.8%) and 111 (33.7%) of the respondents reported that their family members went to health institutions within three months CBHI users and non-users respectively. The presence of under-five children, CBHI scheme membership, household wealth index, and presence of chronic illness was associated with health care service utilization. Households who were enrolled in CBHI were more likely to use healthcare services than households who were not enrolled. Therefore, health sector leaders and managers in the study

**Table 4. Healthcare utilization and community health insurance enrolment in South Gondar Zone, Northwest Ethiopia, 2021 (n = 658).**

| Variables | Health care utilization | | Chi-square $X^2$ | P-values |
|---|---|---|---|---|
| Insurance (CBHI) status | Yes n (%) | No n (%) | 76.3 | <0.001* |
| Non-enrolled | 111 (33.7) | 218 (66.3) | | |
| Enrolled | 223 (67.8) | 106 (32.2) | | |

*Statistically significant at p<0.05.

area should strengthen their efforts for increasing the enrollment of the community into CBHI.

## Supporting information

**S1 File. SPSS data file of the manuscript.**
(SAV)

**S2 File. Pre-tested questionary for community-based health insurance, healthcare service utilization and associated factors in South Gondar Zone Northwest, Ethiopia, 2021: A comparative cross-sectional study.**
(DOCX)

## Acknowledgments

The authors' heartfelt thank goes to data collectors, supervisors, and South Gondar Zone Health Department for their valuable support throughout our work.

## Author Contributions

**Conceptualization:** Fentaw Teshome Dagnaw, Melkalem Mamuye Azanaw, Aytenew Adamu, Tinsaea Ashagrie, Abdelah Alifnur Mohammed, Hiwot Yisak Dawid, Mulu Tiruneh, Biruk Demissie, Getaneh Atikilt Yemata, Getachew Yideg Yitbarek, Yikeber Abebaw, Habtamu Shimels Hailemeskel.

**Data curation:** Aytenew Adamu, Yikeber Abebaw, Habtamu Shimels Hailemeskel.

**Formal analysis:** Fentaw Teshome Dagnaw, Aytenew Adamu, Tinsaea Ashagrie, Getachew Yideg Yitbarek, Yikeber Abebaw, Habtamu Shimels Hailemeskel.

**Funding acquisition:** Aytenew Adamu.

**Investigation:** Fentaw Teshome Dagnaw.

**Methodology:** Fentaw Teshome Dagnaw, Melkalem Mamuye Azanaw, Tinsaea Ashagrie, Abdelah Alifnur Mohammed, Getaneh Atikilt Yemata, Getachew Yideg Yitbarek, Yikeber Abebaw, Habtamu Shimels Hailemeskel.

**Project administration:** Fentaw Teshome Dagnaw, Aytenew Adamu, Abdelah Alifnur Mohammed, Hiwot Yisak Dawid, Biruk Demissie, Getaneh Atikilt Yemata, Getachew Yideg Yitbarek, Habtamu Shimels Hailemeskel.

**Resources:** Fentaw Teshome Dagnaw, Abdelah Alifnur Mohammed, Habtamu Shimels Hailemeskel.

**Software:** Fentaw Teshome Dagnaw, Melkalem Mamuye Azanaw, Aytenew Adamu, Tinsaea Ashagrie, Hiwot Yisak Dawid, Mulu Tiruneh, Biruk Demissie, Getaneh Atikilt Yemata, Getachew Yideg Yitbarek, Yikeber Abebaw, Habtamu Shimels Hailemeskel.

**Supervision:** Fentaw Teshome Dagnaw, Melkalem Mamuye Azanaw, Mulu Tiruneh, Getachew Yideg Yitbarek, Habtamu Shimels Hailemeskel.

**Validation:** Fentaw Teshome Dagnaw, Aytenew Adamu, Abdelah Alifnur Mohammed, Biruk Demissie, Getachew Yideg Yitbarek, Habtamu Shimels Hailemeskel.

**Visualization:** Fentaw Teshome Dagnaw, Aytenew Adamu, Mulu Tiruneh, Getachew Yideg Yitbarek, Habtamu Shimels Hailemeskel.

**Writing – original draft:** Fentaw Teshome Dagnaw, Melkalem Mamuye Azanaw, Aytenew Adamu, Abdelah Alifnur Mohammed, Biruk Demissie, Getaneh Atikilt Yemata, Getachew Yideg Yitbarek, Habtamu Shimels Hailemeskel.

**Writing – review & editing:** Fentaw Teshome Dagnaw, Melkalem Mamuye Azanaw, Aytenew Adamu, Tinsaea Ashagrie, Abdelah Alifnur Mohammed, Hiwot Yisak Dawid, Mulu Tiruneh, Biruk Demissie, Getaneh Atikilt Yemata, Getachew Yideg Yitbarek, Yikeber Abebaw, Habtamu Shimels Hailemeskel.

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
