## [Decision Letter · Decision Letter 0]

1 May 2022

PONE-D-22-04561Community-based health insurance, healthcare service utilization and associated factors in South Gondar Zone Northwest, Ethiopia, 2020: A comparative cross-sectional study.PLOS ONE

Dear Dr. Dagnaw,

Thank you for submitting your manuscript to PLOS ONE. After careful consideration, we feel that it has merit but does not fully meet PLOS ONE’s publication criteria as it currently stands. Therefore, we invite you to submit a revised version of the manuscript that addresses the points raised during the review process.

I want to congratulate the authors for researching determinants of community-based health insurance in LMICs. Kindly consider the suggestions and rectify/re-explain the methodological points mentioned by the second reviewer. The sampling queries and the results in the result section need to be revisited.

We look forward to receiving your revised manuscript.

Kind regards,

Gopal Ashish Sharma, MBBS, MD

Academic Editor

PLOS ONE

Journal Requirements:

Additional Editor Comments:

Universal Health Coverage's key component is enhanced utilisations amongst the eligible participants. The manuscript submitted that study participants predominantly belong to the poor(47.6%). Additionally, ~ half of the sampled participants could not read and write. The observations are probably self-explanatory to utilisations and major focus areas to be worked upon by the authorities.

The authors could add more to the understanding of the content readers if they could divulge the details about various provisions under CBHI, particularly in the introduction section of the manuscript. Further research into Out of Pocket Expenditures amongst the users of CBHI would further highlight the determinants of utilisations or otherwise.

Reviewers' comments:

Reviewer's Responses to Questions

**Comments to the Author**

1. Is the manuscript technically sound, and do the data support the conclusions?

Reviewer #1: Yes

Reviewer #2: Partly

2. Has the statistical analysis been performed appropriately and rigorously? 

Reviewer #1: Yes

Reviewer #2: Yes

3. Have the authors made all data underlying the findings in their manuscript fully available?

Reviewer #1: Yes

Reviewer #2: No

4. Is the manuscript presented in an intelligible fashion and written in standard English?

Reviewer #1: Yes

Reviewer #2: No

5. Review Comments to the Author

Reviewer #1: Community-based health insurance is an alternative to bad access to healthcare due to financial constraints. This piece of evidence will aid decision-makers towards strengthening health care utilization through insurance. The manuscript has been well-framed, and the study's robustness is quite reflective. The result section requires minor edits (line no. 236-237).

Reviewer #2: The authors are requested to kindly describe the findings in results and discussion section in past tense preferably (For example lines 236-7).

In line 147, the authors are requested to specify the need of specifically mentioning about the elderly population in the inclusion criteria as the same is not reflected in results and discussion.

In line 159, the authors are requested to kindly cite reference for "principal component analysis" as the same has been used to categorize the study participants into rich, medium and poor based on the wealth index.

In the sampling procedure mentioned at line 174, it is not clear that in each household how many members were interviewed. If only one member was interviewed then what was the selection criteria for selecting that household member? If all the members present at the time of interview in a single household, then what was the selection criteria to select a respondent? Could the authors have used KISH method for household survey? In short, how did they eliminate selection bias, and information bias?

Please explain Table 4 in detail. Does the term "enrolled" in line 242 mean users and non users as mentioned in the table 4? The description is not clear with respect to table. It would be better to explain that out of those study participants who had enrolled themselves for the CBHI, how many were using the services and how many were not? And whether, the difference in the utilization of health services was significant or not?

In the discussion section, it is may be advisable to use only proportions rather than using absolute numbers (eg line 253)

The statement mentioned in lines 267-8 regarding respondents with chronic illness and their utilization of health services (compared to what?)

6. PLOS authors have the option to publish the peer review history of their article (what does this mean?). If published, this will include your full peer review and any attached files.

Reviewer #1: No

Reviewer #2: **Yes: **Rahul Gupta

---

## [Author Response · Author response to Decision Letter 0]

15 May 2022

Date: 11th May 2022

Dear Editor-in-chief

RE: MANUSCRIPT NUMBER: PONE-D-22-04561

Thank you for your prompt feedback on our manuscript. We especially appreciate the valuable critique provided by the reviewers. We have appropriately addressed the requested revisions as indicated below in which the text written in green shows the response. As instructed, we have electronically resubmitted a revised manuscript incorporating various revisions as outlined below. We remain available to further edit/revise the manuscript as you may require. In the meantime, we trust that these revisions meet with your approval. 

Yours sincerely, 

Fentaw Teshome

(Corresponding author)

Response to Reviewer 

Journal Requirements:

and https://journals.plos.org/plosone/s/file?id=ba62/PLOSOne_formatting_sample_title_authors_affiliations.pdf

Thank you for the comment. It was corrected as per the comment.

Thank you for the comment. It was corrected as per the comment.

Thank you for the comment. The Data Availability statement was “The data is available from the corresponding author and will be provided upon a reasonable request.” It was corrected as per the comment.

Thank you for the comment. It was corrected as per the comment.

Thank you for the comment. It was corrected as per the comment.

Additional Editor Comments:

Universal Health Coverage's key component is enhanced utilisations amongst the eligible participants. The manuscript submitted that study participants predominantly belong to the poor (47.6%). Additionally, ~ half of the sampled participants could not read and write. The observations are probably self-explanatory to utilisations and major focus areas to be worked upon by the authorities.

Thank you for the comment.

The authors could add more to the understanding of the content readers if they could divulge the details about various provisions under CBHI, particularly in the introduction section of the manuscript. Further research into Out of Pocket Expenditures amongst the users of CBHI would further highlight the determinants of utilisations or otherwise.

Thank you for the comment. The aim of this research was majorly to compare and identify factors of health care utilization among CBHI enrolled and non-enrolled households (usually Out of Pocket Expenditures).

Review Comments to the Author

Reviewer #1: Community-based health insurance is an alternative to bad access to healthcare due to financial constraints. This piece of evidence will aid decision-makers towards strengthening health care utilization through insurance. The manuscript has been well-framed, and the study's robustness is quite reflective. The result section requires minor edits (line no. 236-237).

Thank you for your constructive comment. It was corrected as per the reviewer's comment.

Reviewer #2: The authors are requested to kindly describe the findings in results and discussion section in past tense preferably (For example lines 236-7).

Thank you, we appreciate the comment. It was corrected as per the reviewer's comment.

In line 147, the authors are requested to specify the need of specifically mentioning about the elderly population in the inclusion criteria as the same is not reflected in results and discussion.

The need of mentioning the elderly population in the inclusion criteria was that the elderly population tends to utilize healthcare since they are prone to age-related diseases. This was clearly stated in the result and discussion part of the manuscript. The reviewer can re-visit it in the revised manuscript.

In line 159, the authors are requested to kindly cite reference for "principal component analysis" as the same has been used to categorize the study participants into rich, medium and poor based on the wealth index.

Thank you, we appreciate the comment. It was corrected as per the reviewer's comment.

In the sampling procedure mentioned at line 174, it is not clear that in each household how many members were interviewed. If only one member was interviewed then what was the selection criteria for selecting that household member? If all the members present at the time of interview in a single household, then what was the selection criteria to select a respondent? Could the authors have used KISH method for household survey? In short, how did they eliminate selection bias, and information bias?

One member of the household who was the household head was interviewed. We thought that the household head represents the household members and represents the target population. The household head is the one who had better information about CBHI membership and health care utilization. Due to this reason, we didn’t use the KISH method for this study. Since our study was a comparative cross-sectional study, it may not be prone to selection bias and information bias, unlike follow-up studies. 

Please explain Table 4 in detail. Does the term "enrolled" in line 242 mean users and non users as mentioned in the table 4? The description is not clear with respect to table. It would be better to explain that out of those study participants who had enrolled themselves for the CBHI, how many were using the services and how many were not? And whether, the difference in the utilization of health services was significant or not?

Thank you, we appreciate the comment. It was corrected as per the reviewer's comment.

In the discussion section, it is may be advisable to use only proportions rather than using absolute numbers (eg line 253)

Thank you, we appreciate the comment. It was corrected as per the reviewer's comment.

The statement mentioned in lines 267-8 regarding respondents with chronic illness and their utilization of health services (compared to what?)

Thank you for the question. It was compared with respondents who had no chronic illness and it was corrected as per the reviewer's comment.

---

## [Decision Letter · Decision Letter 1]

17 Jun 2022

Community-based health insurance, healthcare service utilization and associated factors in South Gondar Zone Northwest, Ethiopia, 2021: A comparative cross-sectional study.

PONE-D-22-04561R1

Dear Dr. Fentaw Teshome Dagnaw,

We’re pleased to inform you that your manuscript has been judged scientifically suitable for publication and will be formally accepted for publication once it meets all outstanding technical requirements.

Kind regards,

Gopal Ashish Sharma, MBBS, MD

Academic Editor

PLOS ONE

Additional Editor Comments (optional):

Reviewers' comments:

Reviewer's Responses to Questions

**Comments to the Author**

1. If the authors have adequately addressed your comments raised in a previous round of review and you feel that this manuscript is now acceptable for publication, you may indicate that here to bypass the “Comments to the Author” section, enter your conflict of interest statement in the “Confidential to Editor” section, and submit your "Accept" recommendation.

Reviewer #2: All comments have been addressed

2. Is the manuscript technically sound, and do the data support the conclusions?

Reviewer #2: Yes

3. Has the statistical analysis been performed appropriately and rigorously? 

Reviewer #2: Yes

4. Have the authors made all data underlying the findings in their manuscript fully available?

Reviewer #2: Yes

5. Is the manuscript presented in an intelligible fashion and written in standard English?

Reviewer #2: Yes

6. Review Comments to the Author

Reviewer #2: All the comments mentioned in my previous review have been addressed satisfactorily. There are no further comments from my end.

7. PLOS authors have the option to publish the peer review history of their article (what does this mean?). If published, this will include your full peer review and any attached files.

Reviewer #2: **Yes: **Rahul Gupta

---

## [Editor Report · Acceptance letter]

21 Jun 2022

PONE-D-22-04561R1 

Community-based health insurance, healthcare service utilization and associated factors in South Gondar Zone Northwest, Ethiopia, 2021: A comparative cross-sectional study. 

Dear Dr. Dagnaw:

I'm pleased to inform you that your manuscript has been deemed suitable for publication in PLOS ONE. Congratulations! Your manuscript is now with our production department. 

Kind regards, 

on behalf of

Dr. Gopal Ashish Sharma 

Academic Editor

PLOS ONE